# What Lithuanian First-Graders Eat: Results of a 15-Year Semi-Longitudinal, Cross-Sectional Surveillance Study

**DOI:** 10.3390/nu16121970

**Published:** 2024-06-20

**Authors:** Ausra Petrauskiene, Silvija Daugelaite, Aurelija Salomskiene, Vita Speckauskiene

**Affiliations:** 1Faculty of Public Health, Lithuanian University of Health Sciences, 44307 Kaunas, Lithuania; ausra.petrauskiene@lsmu.lt (A.P.); vita.speckauskiene@lsmu.lt (V.S.); 2Faculty of Medicine, Lithuanian University of Health Sciences, 44307 Kaunas, Lithuania; 3Institute of Biological Systems and Genetic Research, Lithuanian University of Health Sciences, 44307 Kaunas, Lithuania; aurelija.salomskiene@lsmu.lt

**Keywords:** first-grade students, dietary habits, changes

## Abstract

This article presents the dietary habits of Lithuanian first-grade (7–8-year-old) students over a 15-year surveillance period to understand the trends and changes in their nutrition patterns. The presented data were collected from three study rounds of the Lithuanian Growth Surveillance Study conducted between 2008 and 2023, with a total sample of 11,594 first-grade students from all 10 counties of Lithuania. The main findings reveal significant shifts in breakfast consumption, with an increase in daily breakfast intake observed over the surveillance period. Conversely, the consumption of cereal porridge showed a notable decrease, particularly in the frequency of consumption. Positive changes were noted in the consumption of vegetables and fresh fruits, indicating an improvement in dietary quality. Also, a concerning trend of declining consumption of certain nutritious food groups like fish and dairy products is identified, whereas the consumption of sugary beverages is low. These findings underscore the importance of ongoing efforts to promote healthier eating habits among school-age children in Lithuania. Addressing these trends requires a multifaceted approach involving education, policy changes, and community-based interventions to ensure the long-term health and well-being of children.

## 1. Introduction

Children’s diet plays a crucial role in their health, growth, development, mental and physical performance, and emotional well-being [1]. To understand nutritional needs and their potential impact on health, healthy eating patterns can be simply described as those rich in health-promoting foods, including plant-based foods, fresh vegetables and fruits, nuts, sources of omega-3 fatty acids, and products low in saturated fats and trans-fats, animal-derived proteins, and added/refined sugars [2]. Carbohydrates serve as the primary source of energy, with the highest amounts found in grains, fruits, legumes, and vegetables [3]. Fresh vegetables and fruits provide energy and dietary fibre, promoting a feeling of fullness and positively affecting digestive tract activity, cholesterol levels, and glycaemic control [4]. Consumption of vegetables and fruits is linked to a reduced risk of cardiovascular diseases, chronic obstructive pulmonary disease, lung cancer, and metabolic syndrome [5,6,7]. While fats are primary structural components of cell membranes and sources of cellular energy, their consumption should be moderate. Valuable unsaturated fats are found in various food products such as fish, avocado, nuts and seeds, and many plant-based oils, whereas animal products and certain plant-based oils contain a higher proportion of saturated fats [8]. Dietary proteins and amino acids are essential building materials for cells, obtained from both animal and plant-based sources [9]. Micronutrients are crucial for normal growth, metabolism, physiological function, and cell integrity maintenance. Deficiency of vitamins and minerals can lead to chronic metabolic disorders associated with cellular ageing and late-onset diseases [10].

Dietary habits gained in childhood often persist into adulthood [11], making it harder to change unhealthy eating patterns later in life. Inappropriate and unbalanced diets among children are significant societal issues worldwide [12,13]. Despite increasing attention to nutrition in both scientific and public domains in recent decades, research indicates that children’s diets still do not meet healthy eating recommendations [3,14]. An international comparison of the dietary habits of Lithuanian first-graders reveals that our country’s indicators often fall below average when it comes to the consumption of fruits and vegetables, breakfast frequency, and intake of fish and legume products [15]. Parents can greatly influence children’s dietary habits from infancy to adolescence, so it is essential to highlight the necessity of parental knowledge and skills regarding food to ensure healthy child nutrition [16]. The first-grade age group is unique in that children, upon starting school, gain greater independence in choosing food products while eating at school, yet parents still play a significant role [17]. The findings of this study could contribute to informing parents about the importance of nutrition and to actively involving them in the cultivation of healthy dietary habits for children.

National, World Health Organization, and European Union strategic documents advocate nutrition research and assessment, solving related health problems, and implementing active measures promoting healthy eating habits among the population. These aim to reduce the prevalence of chronic non-communicable diseases and their risk factors, while finding ways to ensure greater supply and consumption of healthy foods [18,19,20].

It is important for children to eat healthily to avoid the consequences of improper nutrition (underweight, micronutrient deficiency, obesity). Equally important is to study and monitor how children’s eating habits change over time. Numerous cross-sectional studies evaluating the dietary habits of school-age children have been conducted in Lithuania. However, there is a lack of studies aimed at understanding how the dietary habits of Lithuanian children change over time, especially since 2011, following the entry into force of the order of the Minister of Health, “On the Organization of Children’s Nutrition”, which approved the description of the organisation of nutrition in preschool, general education schools, children’s recreation camps, and childcare institutions [21].

Drawing attention to the critical role of nutrition in childhood and deteriorating children’s health, this article aims to assess and describe certain dietary habits and their changes among first-grade students in Lithuania between 2008 and 2023.

## 2. Materials and Methods

This publication presents the results from the Lithuanian Child Growth Surveillance Study, which applies a semi-longitudinal design, meaning that a new cross-sectional sample of first-grade (7–8-year-old) children was selected on each study round [22]. The study was periodically carried out from 2008 as part of the WHO European Childhood Obesity Surveillance Initiative (WHO European COSI) project initiated by the WHO European Regional Office, and was conducted according to the protocol and methodology developed by the WHO and participating member states [22,23]. The surveillance study was performed by researchers from the Health Research Institute at the Faculty of Public Health of the Lithuanian University of Health Sciences with the permission of the Lithuanian Bioethics Committee (No. 16 of 13 March 2008) and its extensions (No. 6B-10-02 of 4 January 2009; No. 6B-13-17 of 9 January 2013; No. 6B-17-211 of 20 October 2017; No. 6B-19-45 of 12 March 2019; and No. 6B-23-54 of 24 March 2023).

Each study round, a national representative sample (*n* = 5800) of first-graders in Lithuania was calculated based on the number of seven-year-old children in each county (according to data from the Lithuanian Department of Statistics, now the State Data Agency) [24]. Using cluster sampling methods, two-stage cluster sampling was applied using the primary school as the primary sampling unit and school classes as the secondary sampling unit. Educational institutions in ten of the country’s counties were selected from the list of Lithuanian schools where children of primary school age are enrolled. With the consent of the school principals, all first-grade classes in the selected schools were invited to participate in the study. The research in educational institutions was conducted by trained and standardised researchers. The study consisted of two stages: During the first visit to the selected schools, envelopes containing the Informed Consent Form and the Family Record Form were distributed to the parents of first-graders. During the second visit, envelopes containing completed questionnaires and signed Informed Consent Forms were collected. A Family Record Form was used to gather information on a voluntary basis on simple indicators of the children’s family socioeconomic factors, physical activity/inactivity patterns, and dietary intake such as the child’s frequency of eating breakfast in a typical week and eating certain food items in a typical week (fresh fruit and vegetables (excluding potatoes); 100% fruit juice, soft drinks containing sugar, and diet or light soft drinks; low-fat/semi-skimmed milk, whole-fat milk, flavoured milk, cheese, and yoghurt/quark/other dairy products; meat, fish, and legumes (e.g., beans, lentils, chickpeas); savoury snacks such as potato chips, corn chips, popcorn, and peanuts; sweet snacks such as candy bars or chocolate, biscuits, cake, and doughnuts; and foods such as pizza, French fries, fried potatoes, hamburger, and sausage [22]). In this study, we analysed the data on having breakfast and the frequency of consumption of various foods during a period of 15 years. In 2008, the study involved 155 schools, with 4375 parents of first-graders surveyed (response rate: 72.3%); in 2016, it involved 102 schools with 3864 parents surveyed (response rate: 69.8%); and in 2023, 94 schools were involved with 3355 parents surveyed (response rate: 57.8%).

Statistical analysis of the research data was performed using IBM SPSS Statistics 29.0 software. Qualitative variables were described by frequency and relative frequency percentage (%). The Chi-square (χ^2^) test was applied to determine the homogeneity of distributions in the three study groups, and the Bonferroni z criterion was used to identify statistically significant differences between subgroups. A significance level of α = 0.05 was chosen, and a difference between compared groups was considered statistically significant when *p* < 0.05.

## 3. Results

The aim of this paper is to evaluate the nutrition habits of Lithuanian first-grade students and their changes during a period of 15 years. The results were collected during six study rounds. In this article, we focus on the 2008, 2016, and 2023 study rounds. A total of 11,594 first-grade children from all 10 counties of Lithuania were included in the analysis.

### 3.1. Breakfast Consumption

During the 15 years of the surveillance study, the rate of first-grade Lithuanian children having breakfast every day increased and the proportion of children never consuming breakfast decreased statistically significantly (Figure 1).

### 3.2. Cereal/Grain Porridge Consumption

Concerning grain products, two new questions were added to the questionnaire in 2023: cereals with and without sugar. The frequency of consumption of cereal with sugar was as follows: never or less than once a week: 1762 (53.3%), some days (1–3 d./w.): 1078 (32.6%), most days (4–6 d./w.): 310 (9.4%), and every day: 159 (4.8%). However, the consumption of cereal without sugar in 2023 was less frequent (than with sugar) and statistically significant (χ^2^ = 362.4, df = 3, *p* < 0.001) in all consumption groups: never or less than once a week: 2477 (75.2%), some days (1–3 d./w.): 632 (19.2%), most days (4–6 d./w.): 120 (3.6%), and every day: 63 (1.9%).

Figure 2 indicates a statistically significantly decreased rate of first-grade children having cereals in the most days and every day groups compared by year. The number of children who never consume cereal increased by 1.5 times in 2023 compared to earlier years.

### 3.3. Vegetable and Fresh Fruit Consumption

Positive changes in consuming vegetables and fresh fruits were observed in the year 2023 as the rate of children having these products every day statistically significantly increased and the rate of children never eating vegetables decreased significantly (Figure 3).

### 3.4. Meat and Fish Consumption

The rates of meat consumption statistically significantly increased in the most days and every day groups among first-graders compared by year (Figure 4). However, fish consumption rates are disappointing—more than half of the children do not consume fish at all while one-third of the children eat fish 1–3 days per week.

### 3.5. Milk Consumption

The proportion of children consuming semi-skimmed and whole-fat milk every day decreased statistically significantly over the surveillance period (Figure 5). There was a significant increase in children consuming semi-skimmed milk most days or some days a week. The consumption of whole-fat and flavoured milk statistically significantly decreased in each study year.

### 3.6. Dairy Product Consumption

Declining trends in the consumption of other dairy products were also observed (Figure 6). The everyday consumption of cheese, yoghurt, curd, and other dairy products among Lithuanian first-graders was the highest in 2008 and significantly decreased in 2023. The percentage of children who consumed dairy products on some days per week or did not consume these products at all increased significantly.

### 3.7. Juice and Soft Drink Consumption

Consumption of natural juice is also decreasing: in the past years we observed a significant increase in children never consuming fruit juice and a significant decrease in fruit juice consumption (Figure 7). The same tendency to decrease was observed in the consumption of soft drinks containing sugar as the rates were lower and statistically significant in the never and some days groups.

The consumption of soft drinks without sugar (e.g., Light Coca-Cola) was less popular among Lithuanian children: the comparison between years was statistically significant, showing a decrease from 2008 to 2016 and to 2023 (χ^2^ = 196.1, df = 6, *p* < 0.001). The rate for the year 2023 in the never group was 83.7%, some days—12.8%, and the rest—less than 4%, which included the most days and every day groups.

## 4. Discussion

In this publication, we aimed to assess and describe certain dietary habits and their changes among representative samples of first-grade students in Lithuania, as well as to present some results from the Lithuanian Child Growth Surveillance Study of three survey rounds from 2008 to 2023.

All children should eat breakfast every day. Comparing the data from 2008 to 2016 and to 2023, it is seen that during the study period the number of first-graders skipping breakfast tended to decrease by more than two times, and the number of children eating breakfast daily increased significantly. However, it was found that almost one-third of the participants still skip breakfast daily. Other scientific studies from our country indicate that a considerable proportion of children still skip breakfast daily [25,26,27,28,29], but our surveillance study shows that over the years the situation has been improving. The European COSI average of children eating breakfast every day was 75% in 2018–2020. The levels of daily breakfast consumption ranged from 94% in Portugal and Denmark to 44% in Armenia and 49% in Greece [12].

In the Lithuanian Healthy and Sustainable Eating Recommendations [30], it is advised to eat vegetables and fruits several times a day. According to the rules for the implementation of the programme to promote the consumption of fruits and vegetables, as well as dairy products in educational institutions for children approved by the Minister of Agriculture of the Republic of Lithuania on September 21, 2017, the aim is that 70% of children attending educational institutions consume fruits and vegetables several times a day [31].

Although the consumption of fruits and vegetables significantly increased in 2023 compared to 2008, according to the results of the surveillance study, still less than half of our first-grade children consume vegetables and fruits daily. The problem of insufficient consumption of vegetables, fruits, and berries has been pointed out by other Lithuanian scientists as well [25,26,27,28,29]. Comparing the 2018–2020 results from other European countries, the proportion of children consuming fruit and vegetables daily was highest in Portugal, Ireland, and Denmark (≥60% and 57%) and lowest in Georgia, Spain, and Latvia (23%, 27%, and 13%, respectively); the proportion of Lithuanian children consuming fruit and vegetables every day was 32% and 27%, respectively, in the fifth round [12] and more than two-fifths in 2023.

Meat and its products consist of biologically active substances—essential amino acids, fat-soluble and B-group vitamins, and minerals (macronutrients and micronutrients)—but fatty meat and high-calorie meat products should be consumed moderately [32]. The data from the study showed a trending increase in the proportion of children consuming meat daily and 4–6 times a week and a decrease in those consuming it 1–3 times a week.

Fish is a protein-rich and low-calorie food, easily digestible, and rich in the fat-soluble vitamins A, D, and E, as well as minerals such as copper, zinc, selenium, fluoride, and iron [33]. Fish is particularly rich in polyunsaturated omega-3 fatty acids, which are beneficial for vision, strengthening the immune and nervous systems, and helping to maintain normal heart function [34,35]. It is recommended to consume fish and its products at least 2–3 times a week; they can be a suitable alternative to red meat. According to the data from our study, there has been a significant increase in those who never eat fish, while most respondents who ate fish consumed it less than once a week, and only slightly more than a third of first-graders’ fish consumption habits in 2023 met the recommendations.

Milk and dairy products are rich in proteins containing essential amino acids, as well as fats, carbohydrates, fat-soluble vitamins, and minerals [36,37]. They are rich in calcium, which is especially necessary for children as it forms their bones and teeth and supports the nervous system and the activity of internal and external secretion glands, blood clotting processes, and so on [38,39,40]. Comparing the data from different stages of the study, it is seen that the consumption of milk and its products is tending to decrease: the proportion of first-graders who never consume semi-skimmed and natural milk has almost doubled, and especially decreased is the proportion of first-graders who consume yoghurt, curd, or other dairy products daily.

To ensure the formation of a health-promoting diet in schools, the order of the Minister of Health, “On the Organization of Children’s Nutrition” [21], aimed to create conditions for health-friendly nutrition, ensure the best food safety and quality, meet children’s physiological needs for food nutrients, and develop healthy eating skills. Vegetables and fruits should be served every day. Hot food must be prepared and served on the same calendar day. Hot lunch should consist of high-protein products (meat, poultry, fish, eggs, legumes, milk and dairy products) and high-carbohydrate products. Whole-grain or partially whole-grain products are preferred. The most diverse cereals or cereal flakes are recommended (oats, buckwheat, rice, wheat, barley, pearl barley, corn, etc.). Cereals or cereal flakes (except polished rice) are recommended at least three times per week and breakfast cereals no more than once per week [21]. As can be seen from our survey, the consumption of breakfast cereals is quite low among first-graders, but despite the recommendations children do not eat grain porridge.

School canteens are prohibited from providing children with unhealthy products such as potato, corn, or other chips; products cooked in fat; and sweets, ice cream, sodas, energy drinks, and the like. The public health specialist, who provides health care at school, assesses the compliance of the organisation of children’s meals with the requirements [21]. While analysing the changes in the consumption of sugary soft drinks, it was found that there was a significant increase in the proportion of children consuming these drinks 1–3 times a week or less frequently and those never consuming them. In 2023, the proportion of children consuming 100% fruit juice less than once a week and those never drinking them almost doubled. According to parental data, most first-graders consume salty snacks and sweets less than once a week and 1–3 times a week. It can be assumed that parents are aware of the recommendations for nutrition in educational institutions and the moderate consumption of unhealthy products.

It is known that animal-derived proteins are better absorbed, and their amino acids better meet the human body’s needs. However, observing the decrease in the consumption of milk and its products and fish, the fact that a fourth of the respondents do not eat legumes at all, and only some of the first-graders consume vegetables and fruits daily, raises a reasonable question: are the physiological needs of growing children fully met, especially considering that the health of children is worsening and the prevalence of overweight and obesity and the risk of non-communicable diseases in adult life are growing? Healthy and balanced nutrition is crucial for children’s growth and development. It has been found that children who eat an unbalanced diet are often shorter than their peers, get sick more often, and perform worse at school. It is also claimed that poor nutrition in childhood and adolescence affects maturation processes, brain development, motor skill development, productivity, susceptibility to illness, and overall health. Improper nutrition in childhood is associated with a higher risk of breast, ovarian, uterine, kidney, and colon cancers, as well as heart and vascular diseases in later life [4,41,42,43].

In conclusion, the findings from this study provide valuable insights into the evolving nutrition habits of Lithuanian first-grade children over a 15-year period. While there have been some positive changes, such as increased breakfast consumption and higher intake of vegetables and fruits, there are also concerning trends, including declining consumption of certain nutritious food groups. Dietary habits formed in childhood often last a lifetime, and not all first-grade children’s dietary habits meet healthy eating recommendations. Addressing these trends will require a multifaceted approach involving the education of parents and children, policy changes, and community-based interventions to promote healthier eating habits and improve the overall nutritional status of children in Lithuania.

## 5. Conclusions

Over the 15-year surveillance period, notable positive and negative changes emerged in the dietary habits of first-graders. The frequency of daily breakfast consumption increased significantly, while the proportion of children skipping breakfast decreased. Similarly, there was a significant increase in the daily intake of vegetables and fresh fruits among first-graders and an increase in the proportion of children consuming meat daily and 4–6 times a week. There was a notable decline in sugary drink consumption, with fewer children consuming fruit juice and soft drinks. The consumption of legumes remained low. Milk and dairy product consumption showed a significant decrease, particularly in daily and weekly consumption frequencies. Fish consumption also decreased significantly, with a notable increase in the proportion of children who never eat fish. Additionally, there was a notable decline in daily and weekly consumption of cereal porridge among first-graders.

## Figures and Tables

**Figure 1 nutrients-16-01970-f001:**
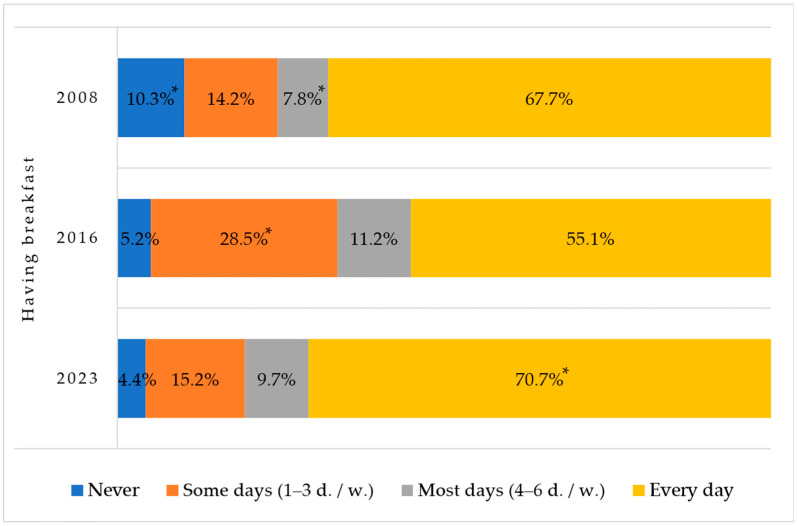
Rates of having breakfast among Lithuanian first-graders in 2008, 2016, and 2023. * Indicates statistically significant differences in the lowest or highest rates of breakfast consumption within each group of children when comparing different years (χ^2^ = 503.9, df = 6, *p* < 0.001).

**Figure 2 nutrients-16-01970-f002:**
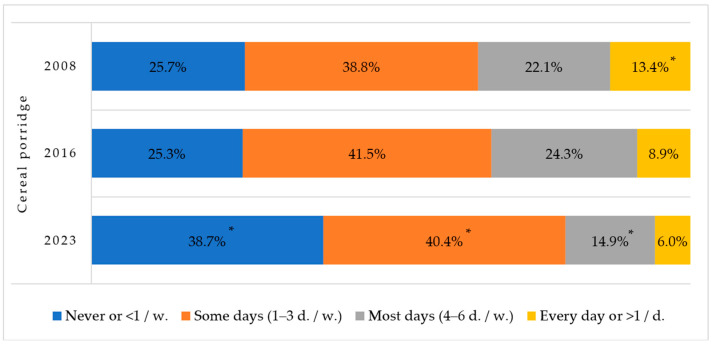
Rates of consuming cereal porridge among Lithuanian first-graders in 2008, 2016, and 2023. * Indicates statistically significant differences in the lowest or highest rates of cereal porridge consumption within each group when comparing between years (χ^2^ = 334.3, df = 6, *p* < 0.001).

**Figure 3 nutrients-16-01970-f003:**
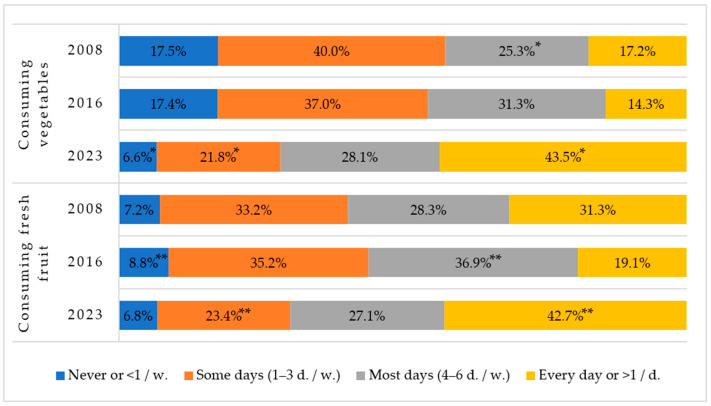
Rates of consuming vegetables and fresh fruits among Lithuanian first-graders between 2008, 2016 and 2023. * Indicates statistically significant differences in the lowest or highest rates of vegetable consumption within each group when comparing between years (χ^2^ = 1162.9, df = 6, *p* < 0.001). ** Indicates statistically significant differences in the lowest or highest rates of fresh fruit consumption within each group when comparing between years (χ^2^ = 477.6, df = 6, *p* < 0.001).

**Figure 4 nutrients-16-01970-f004:**
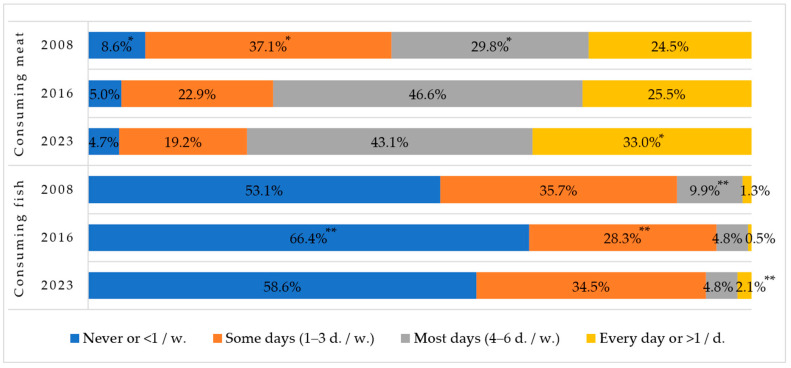
Rates of consuming meat and fish among Lithuanian first-graders in 2008, 2016, and 2023. * Indicates statistically significant differences in the lowest or highest rates of meat consumption within each group when comparing between years (χ^2^ = 532.2, df = 6, *p* < 0.001). ** Indicates statistically significant differences in the lowest or highest rates of fish consumption within each group when comparing between years (χ^2^ = 230.8, df = 6, *p* < 0.001).

**Figure 5 nutrients-16-01970-f005:**
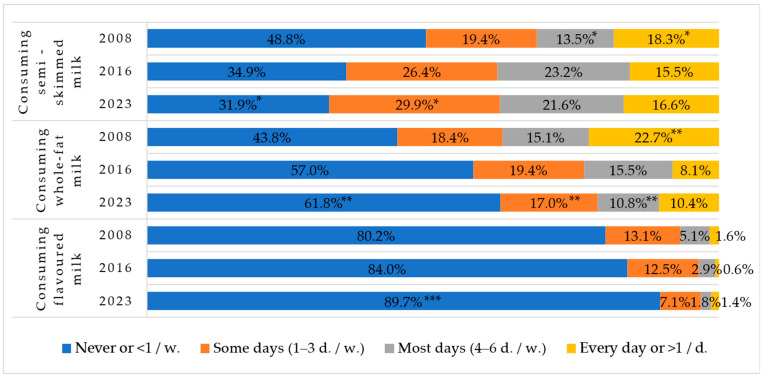
Rates of consuming semi-skimmed, whole-fat, and flavoured milk among Lithuanian first-graders in 2008, 2016, and 2023. * Indicates statistically significant differences in the lowest or highest rates of semi-skimmed milk consumption within each group when comparing between years (χ^2^ = 367.3, df = 6, *p* < 0.001). ** Indicates statistically significant differences in the lowest or highest rates of whole-fat milk consumption within each group when comparing between years (χ^2^ = 506.1, df = 6, *p* < 0.001). *** Indicates statistically significant differences in the highest rates of flavoured milk consumption within each group when comparing between years (χ^2^ = 169.1, df = 6, *p* < 0.001).

**Figure 6 nutrients-16-01970-f006:**
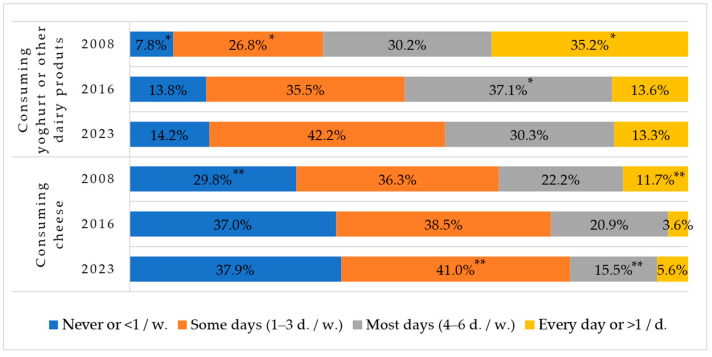
Rates of consuming cheese, yoghurt, or other dairy products among Lithuanian first-graders in 2008, 2016, and 2023. * Indicates statistically significant differences in the lowest or highest rates of yoghurt or other dairy product consumption within each group when comparing between years (χ^2^ = 289.1, df = 6, *p* < 0.001). ** Indicates statistically significant differences in the lowest or highest rates of cheese consumption within each group when comparing between years (χ^2^ = 828.1, df = 6, *p* < 0.001).

**Figure 7 nutrients-16-01970-f007:**
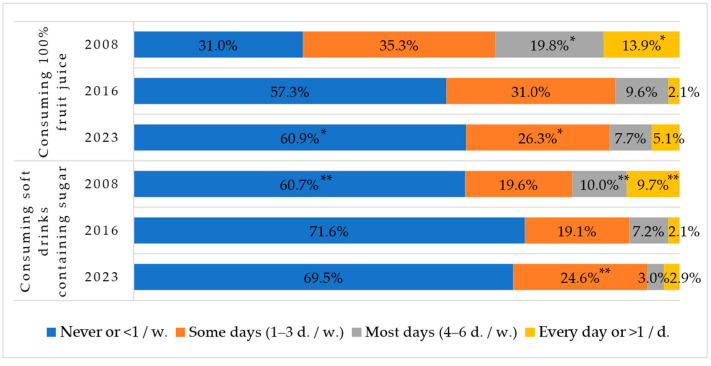
Rates of consuming 100% fruit juice and soft drinks containing sugar among Lithuanian first-graders in 2008, 2016, and 2023. * Indicates statistically significant differences in the lowest or highest rates of 100% fruit juice consumption within each group when comparing between years (χ^2^ = 1124.1, df = 6, *p* < 0.001). ** Indicates statistically significant differences in the lowest or highest rates of soft drinks containing sugar consumption within each group when comparing between years (χ^2^ = 461.6, df = 6, *p* < 0.001).

## Data Availability

The original contributions presented in the study are included in the article, further inquiries can be directed to the corresponding author.

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
