# Peer review of "What Lithuanian First-Graders Eat: Results of a 15-Year Semi-Longitudinal, Cross-Sectional Surveillance Study"

_nutrients, 2024, doi:10.3390/nu16121970_

Round 1
Reviewer 1 Report
Comments and Suggestions for Authors
The present manuscript covers an interesting topic of changes in dietary intake in Lithuanian students within 15 years. Unfortunately, the applied methodology is not sufficiently described, therefore, it is not clear how the data on children’s dietary intake was collected. Another drawback of the manuscript is the lack of proper references in some parts of the Introduction and Discussion (generally there are only 27 articles cited in this manuscript which doesn’t seem to be a lot).
Introduction
-The second paragraph is quite robust, but it lacks some more references (currently there are only 3 cited articles).
-It must be clarified what age first-grade age group encompasses. This information appears in the Results but it must be indicated earlier in the manuscript.
Materials and Methods
-Once again, It must be indicated what age first-grade age group encompasses.
-Row 109: “The family questionnaire primarily comprised closed questions […]” – what kind of questions were asked?
The applied methodology is poorly described. For example, how did you collect data on children’s dietary intake? Did you use a 3-day dietary record or some food frequency questionnaires? Were this questionnaire/ these questionnaires validated to be used in a Lithuanian population?
Results
-What was the reason for assessing the frequency of cereal porridge consumption instead of the frequency of overall cereal products?
Discussion
-To enrich the discussion, it would be valuable to present your data compared to data from any other European country. Are there similar longitudinal studies showing how the dietary habits of first-grade students changed in other countries?
-Rows 250-253 – Please provide a proper reference.
-Rows 256-261 – Please provide a proper reference.
-Rows 267-271 – Please provide a proper reference.
- Rows 298-299: “[…] especially considering that we often hear about worsening health indicators among children, decreasing suitability of young people for military service” – I don’t think that possible decreased suitability of young people for military service is the biggest concern due to worsening health indicators. The mention of the risk of non-communicable diseases in adult life would be appropriate.
Conclusion
-Taking into account received data, what actions should be taken to promote more healthy eating habits in a group of first-graders and minimize improper habits?
Comments on the Quality of English LanguageI have no major reservations
Author Response
On behalf of all authors, I would like to express our sincere gratitude to the reviewer for insightful comments and constructive feedback, which greatly contributed to improving the quality of our manuscript. Your expertise and thoughtful review are deeply appreciated.
Introduction
- We added some more references (by WHO, UNICEF, Mahmood L. et al, Bartkeviciute R. et al.) to the second paragraph.
- We clarified the age of first-graders.
Materials and Methods
- The age group was indicated in the Abstract and in the chapter Materials and Methods (row 13 and 96).
- We corrected the description of food-related questions. Mainly, the Food Frequency Questionnaire was analysed from the Family Record Form. This form included questions on postnatal factors, complementary feeding, physical activity, family's socio-economical factors and ect. Questions were partly based on the questionnaire used in the 2001/2002 round of the Health Behaviour in School-aged Children Study. We also added more information on the Family Record Form (row 121-130).
Results
- Traditionally in Lithuania we consume porridges from various grains, while breakfast cereals are not recommended to consume because usually they contain sugar. The consumption of breakfast cereals in the questionnaire of 2023 was separated into two questions: cereals with and without sugar. The frequency of consumption of grains porridge was the third question on grains.
Discusion
- We compared our data with the data of COSI member states from 2018-2020 as the same methodology and questionnaire was used.
- References are provided on various products consumption recommendations.
We are grateful for your suggestion about the risk of non-communicable diseases.
Conclusions
- We have written conclusions following from our data. Actions which should be taken to promote more healthy eating habits are provided at the end of the Discussion chapter (rows 369-373).
Reviewer 2 Report
Comments and Suggestions for Authors
The paper is an honest illustration of some eating habits relevant to children's future health. The manuscript is complete, but I have some questions to address.
The study is defined as semi-longitudinal, but repeated observation of participants doesn't seem to be done. Please explain better. Otherwise, I suggest removing "semi-longitudinal" from the title.
Did the parents consider the consumption at school, including the frequency of foods promoted by the Ministry of Agriculture program? How were they informed about the consumption at school?
Lines 87-91: "The surveillance is periodically carried out every three years participating in the WHO European Childhood Obesity Surveillance Initiative (WHO European COSI) project initiated by the WHO European Regional Office, and it is conducted according to the protocol and methodology developed by the WHO and participating countries [17,18]."
It is unclear whether the work was supported by the WHO or if it is an initiative following the method but not under the WHO umbrella.
Lines 286-290: "In 2023 the proportion of children consuming 100% fruit juice less than once a week and those who never drink them almost doubled. According to parental data, most first-graders consume salty snacks and sweets less than once a week and 1-3 times a week. It can be assumed that parents are aware of the recommendations for nutrition in educational institutions and some of them consider the moderate consumption of health-unfriendly products."
Does the parents' awareness regard either "salty snack and sweets" or "100% fruit juice" or both?
Moreover, why can it be assumed?
Line 296: Does "every fourth of the respondents does not eat legumes at all" mean that children do not eat peanuts or something similar?
Lines 301-302: "It has been found that children who eat unreasonably are often shorter than their peers, get sick more often, and perform worse at school"
What do the Authors mean by "unreasonably"?
Finally, do the Authors think they should start a longitudinal study or at least a follow-up of the surveyed subjects?
Author Response
On behalf of all authors, I would like to express our sincere gratitude to the reviewer for insightful comments and constructive feedback, which greatly contributed to improving the quality of our manuscript. Your expertise and thoughtful review are deeply appreciated.
The definition of semi-longitudinal study in the article is defined as in the study protocol of WHO European Childhood Obesity Surveillance Initiative. Lithuania participated in this initiative since 2008. 2023 was the sixth round of the surveillance.
Kindergartens and primary schools are encouraged to participate in the Program for Promoting the Consumption of Fruits and Vegetables and Milk and Dairy Products. Ministry of Agriculture of the Republic of Lithuania coordinates this program. Children are receiving fruits, vegetables and milk products for free. Parents are informed about this program.
WHO is coordinating the European Childhood Obesity Surveillance Initiative - COSI and preparing the methodology, giving technical support. Lithuania is participating in the COSI since its planning and very beginning in 2007. We send our data to WHO, joint publications are prepared and published. Every participating country must have their own title for the survey (for example Italians perform the study called OKkio alla SALUTE, in Lithuania we perform Child Growth Surveillance Study).
Parents are informed about nutrition requirements at school in the beginning of the school year. First graders receive hot meal at school for free. Parents are provided with the menu. They are informed about recommended and forbidden products for the children at school.
We provided some clarification in the text about legumes (row 128).
We have changed the word "unreasonably" to "an imbalanced diet" (row 354).
In Lithuania we have already done longitudinal study where the same students were surveyed in the first, third, sixth ad ninth grades all over Lithuania. Final sample was a little bit more than 450 children. The data was published in another article and doctoral dissertation was defeated.